

# Effects of online group exercises for older adults on physical, psychological and social wellbeing: a randomized pilot trial

Marcos Baez[1], Iman Khaghani Far[2], Francisco Ibarra[1], Michela Ferron[3], Daniele Didino[4,5] and Fabio Casati[1]

[1] Department of Information Engineering and Computer Science, University of Trento, Trento, Trentino, Italy
[2] College of Computer & Information Science, Northeastern University, Boston, MA, United States
[3] Fondazione Bruno Kessler, Trento, Trentino, Italy
[4] Department of Psychology, Humboldt-Universität zu Berlin, Berlin, Germany
[5] Department of Economy, Tomsk Polytechnical University, Tomsk, Russia

## ABSTRACT

**Background**. Intervention programs to promote physical activity in older adults, either in group or home settings, have shown equivalent health outcomes but different results when considering adherence. Group-based interventions seem to achieve higher participation in the long-term. However, there are many factors that can make of group exercises a challenging setting for older adults. A major one, due to the heterogeneity of this particular population, is the difference in the level of skills. In this paper we report on the physical, psychological and social wellbeing outcomes of a technology-based intervention that enable online group exercises in older adults with different levels of skills.

**Methods**. A total of 37 older adults between 65 and 87 years old followed a personalized exercise program based on the OTAGO program for fall prevention, for a period of eight weeks. Participants could join online group exercises using a tablet-based application. Participants were assigned either to the Control group, representing the traditional individual home-based training program, or the Social group, representing the online group exercising. Pre- and post- measurements were taken to analyze the physical, psychological and social wellbeing outcomes.

**Results**. After the eight-weeks training program there were improvements in both the Social and Control groups in terms of physical outcomes, given the high level of adherence of both groups. Considering the baseline measures, however, the results suggest that while in the Control group fitter individuals tended to adhere more to the training, this was not the case for the Social group, where the initial level had no effect on adherence. For psychological outcomes there were improvements on both groups, regardless of the application used. There was no significant difference between groups in social wellbeing outcomes, both groups seeing a decrease in loneliness despite the presence of social features in the Social group. However, online social interactions have shown to be correlated to the decrease in loneliness in the Social group.

**Conclusion**. The results indicate that technology-supported online group-exercising which conceals individual differences in physical skills is effective in motivating and enabling individuals who are less fit to train as much as fitter individuals. This not only indicates the feasibility of training together *despite* differences in physical skills but also suggests that online exercise might reduce the effect of skills on adherence

Corresponding author
Marcos Baez, baez@disi.unitn.it

in a social context. However, results from this pilot are limited to a small sample size and therefore are not conclusive. Longer term interventions with more participants are instead recommended to assess impacts on wellbeing and behavior change.

## INTRODUCTION

### Background

Extensive research has documented the association of regular physical activity with positive outcomes in health and wellbeing in later age (*Thibaud et al., 2012*; *Stuart et al., 2008*; *Landi et al., 2010*). Engaging in physical activities reduces the risk of falls (*Thibaud et al., 2012*), slows the progression of degenerative diseases (*Stuart et al., 2008*), and improves cognitive performance and mood in older adults (*Landi et al., 2010*). Conversely, sedentary behaviour is associated with mortality, risk of depression and adverse effects on health and wellbeing in older adults (*Rezende et al., 2014*; *Teychenne, Ball & Salmon, 2010*).

Intervention programs to promote physical activity in older adults, either in group or individual (home) settings, have demonstrated the potential to improve health and functional performance (*El-Khoury et al., 2013*). Both types of intervention have shown equivalent health outcomes (*Freene et al., 2013*) but with different results when considering *adherence*. Group-based interventions seem to achieve higher participation in the long-term (*Van Der Bij, Laurant & Wensing, 2002*), while in the short-term the results are comparable or not conclusive (*Van Der Bij, Laurant & Wensing, 2002*; *Freene et al., 2013*).

The existing evidence for a higher participation to group-based interventions can be explained by the importance of socialization as a motivating factor in physical training (*Phillips, Schneider & Mercer, 2004*; *De Groot & Fagerström, 2011*). As reported by *De Groot & Fagerström (2011)*, older adults do prefer training with others rather than individually. However, there are many factors that can make group exercises a challenging (or infeasible) setting for older adults. A major obstacle, due to the heterogeneity of this broad population, is the big difference in the level of physical abilities between participants: unless the training class is tailored to the needs and abilities of each group, we are likely to see limited effectiveness as well as lack of motivation in performing the exercises (*De Groot & Fagerström, 2011*; *Müller & Khoo, 2014*), which proves itself difficult in heterogeneous groups. This difference in physical abilities, along with the logistic and practical obstacles that make it more difficult, as we age, to regularly attend a gym and perform group exercises, means that, for some adults, home-based individual intervention is the only viable training option.

### Objective

In this paper we report on a technology-based physical intervention that enables older adults with different abilities, and indeed *in spite* of their different abilities, to engage in

group exercises from home while keeping these differences invisible to the group. The intervention is based on the OTAGO Exercise Program for fall prevention (*Gardner et al., 2001*) and supported by a set of applications that allow older adults to follow virtual training sessions via a tablet device under the supervision of a remote Coach.

The study presented in this paper is Part II of the intervention program presented in *Far et al. (2015)* that studied the effect of virtual fitness environments on adherence and social interactions. In this paper we report on the physical, psychological and social wellbeing outcomes related to the intervention. The primary objectives of this study, and of the paper are:

- to investigate if and how online group-exercising and baseline measures of physical, social and psychological wellbeing influence the adherence of older adults to the training program.
- to assess the effectiveness of an OTAGO-based exercise program delivered via an online group-exercising tool - effectiveness measured as the improvements in the physical functions expected by the exercise program.

Additional objectives were to assess the effect of a social (rather than individual) virtual gym that enables to train in group on psychological and social wellbeing outcomes.

## Related work

Previous studies have demonstrated the effectiveness of the OTAGO Exercise Program in reducing falls and fall-related injuries among high risk individuals, and increasing the percentage of older adults who are able to live independently in their community (*Campbell et al., 1997*; *Campbell & Robertson, 2003*). Although this specific program was designed for home-based training, a meta-analysis including other exercise programs for fall prevention found that combining group-based and home-based exercises is a strategy used in several effective trials, thus recommending both options to be available (*Sherrington et al., 2011*).

Technology-based interventions have also demonstrated to be effective in increasing and maintaining physical activity (refer to *Müller & Khoo, 2014* and *Aalbers, Baars & Rikkert, 2011* for systematic reviews). Among the technological components that have been explored we can mention: online newsletters (*Hageman, Walker & Pullen, 2005*), personalized emails (*Ferney et al., 2009*), web-based videos (*Irvine et al., 2013*; *Benavent-Caballer et al., 2015*), tablet applications (*Silveira et al., 2013a*) and video game consoles (*Jorgensen et al., 2013*). However, most of the existing intervention studies have focused on individual training, or provided a social context that was limited to forums or chats (*Aalbers, Baars & Rikkert, 2011*). Even exergames, a type of technology that have traditionally provided more immersive experiences (e.g., via MS Kinect and Nintendo Wii), have not been explored in an online group setting. A systematic review on exergames besides reporting on lack of conclusive results on the improvements in physical functioning, reported only one interventions with exercise performed in pairs, but that required physical presence (*Molina et al., 2014*).

In summary, there are no interventions exploring online group-exercising in home settings, where training programs were tailored to individuals. In this paper, we report

on the feasibility and outcomes of a technology-based intervention in such settings. It complements our previous reports on the same intervention:

- *Baez et al. (2016)*, on the application design and human factors. We describe the design rationale and its evolution, and report on the technology acceptance, usage and usability, and the nature of online interactions.
- *Far et al. (2015)*, on whether the application motivated participants to adhere to the training program, and whether the virtual gym design motivated older adults to train together in a group.

From our previous work we understood that (i) the virtual gym was highly usable and accepted, (ii) it motivated individuals to follow the training program, (iii) and to join the training sessions in groups as opposed to alone. In this paper, however, we focus in understanding how baseline measures of physical, social and psychological wellbeing affect the adherence to the training program, and how this effect is modified by training in an online social setting. We also report on the effectiveness of the training program in terms of physical outcomes, and additionally on social and psychological wellbeing.

## MATERIALS & METHODS

The methods followed in this intervention study have been described in detail in *Far et al. (2015)*. In this section, we elaborate on the aspects related to the specific objectives of this paper.

### Training application

Gymcentral (http://gymcentral.net) is a web and tablet application designed to *enable* and *motivate* older adults of different abilities to participate in group training sessions from home, under the supervision of a human coach. The technology provided by Gymcentral supports the online group-exercising as illustrated in Fig. 1.

The design of the application is based on the metaphor of a *virtual gym*, mimicking the spaces and services found in a real gym. The main features of the service are:

- **Reception**. The entry point to all the services of the gym.
- **Locker room.** As in a real gym, a space where trainees usually meet each other and get ready for the training classes. In the locker room, users can see each other, interact by means of predefined messages (e.g., "Hi, let's go to the classroom"), and invite members who are not online to join.
- **Classroom**. The environment where users have access to the training videos. In this space users are not only able to see the Coach, but also each other as avatars, giving the feeling of training together.
- **Progress report**. It displays the progress of the trainee in the training program by means of a growing garden metaphor.
- **Training schedule**. It displays the training schedule for the week, displaying participation of users in each session, and reminding them of the upcoming sessions.
- **Messaging**. Messaging features allow users to exchange public and private messages. Trainees use this feature to communicate with other trainees and the Coach.

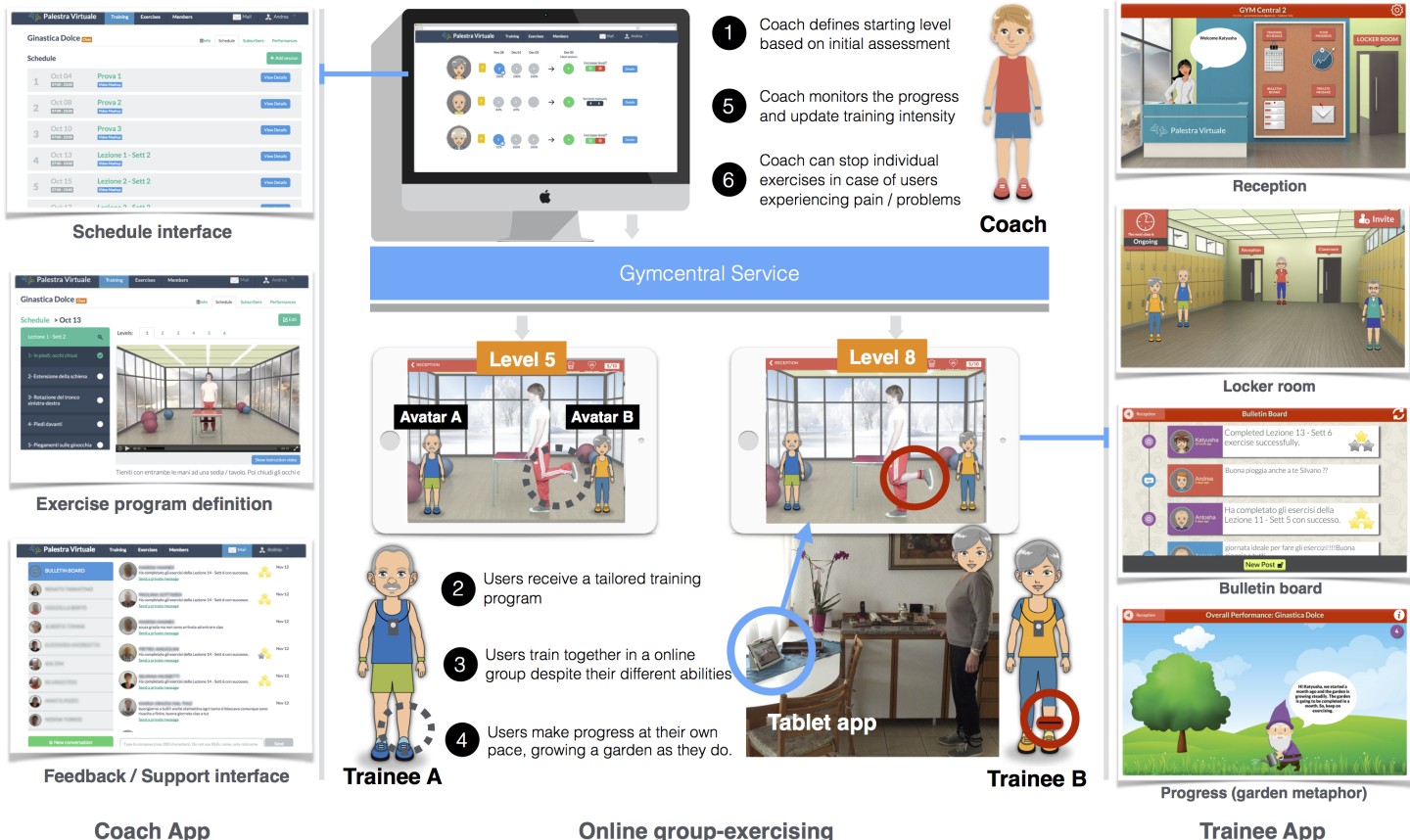

**Figure 1** Online group-exercising workflow (numbers indicate the sequence of activities).

- **Positive and negative reinforcement**. After the completion of a training session, positive or negative reinforcement messages are presented to users depending on the number of exercises completed.

In Gymcentral, personalisation is related to the level of difficulty of the exercise, and it works from two perspectives. On one hand, the user is able to request for an increase in the training level (*level-up*). A coach receives and manages the requests of users (Fig. 1, point 5) and, for each individual case, is able to accept or reject the request, based on user performances. Additionally, each exercise in the training plan can be suspended if necessary (e.g., in case of illness or aches). A key aspect here is that users can train in a virtual group even if each participant is shown different exercise instructions and videos, matching their personalized level of training. Even if users are able to see their training partners exercising while they are in the same training sessions, they are not aware of the difference in training levels. Therefore, trainees can train together despite their different capabilities.

The main difference between Gymcentral and previous tools is in the possibility of following group training programs from home, in a virtual gym, supervised by a remote Coach. A detailed discussion of the design and evolution of the Gymcentral app can be found in *Baez et al. (2016)*.

## Study design

The study followed a framework for the design and evaluation of complex interventions in health settings (*Campbell et al., 2000*). Using a matched random assignment procedure, participants were assigned either an experimental ("social") condition or to a control condition (*McBurney & White, 2009*; *Whitley & Kite, 2013*), considering *age* and *participants' frailty level* as the random assignment variables. The randomization was performed using the statistical software SPSS. The allocation was concealed to the researchers who enrolled and assessed the participants (the Coach and a social scientist). Once baseline measures were taken, a third researcher, using the statistical software SPSS, performed the random assignment (at once) to Control and Social conditions from the pool of 40 subjects. The overall study flow is depicted in Fig. 2.

Participants in the social group were given a version of the Gymcentral Trainee App that included the personalized training program, social environment for group exercising, messaging and persuasion features. Participants were aware that they were exercising together and they could choose to do so. In the control condition, participants received a version of the application that focused only on the home-based program, delivering the personalised training but without social or individual persuasion features. Participants from both groups were offered technology training modules (∼1.5 h each) focusing on operating the tablet, the use of the main applications and the Gymcentral app. The training took place after the pre-measurements.

As part of the study kit, participants received a 10.1 inch Sony Xperia tablet with the assigned version of the application installed, the user guide including the names and telephone numbers of the support team, instructions about the use of the tablet and the assigned application, one pair of ankle weights to perform the exercises and a folder to allow the vertical positioning of the tablet.

Pre- and post- measurement took place before and after the study. The initial measures were collected in three meetings: (i) in the first meeting, demographic information, the *Groningen Frailty Indicator* score (GFI; *Steverink et al., 2001*) and the *Rapid Assessment of Physical Activity Questionnaire* score (RAPA; *Topolski et al., 2006*) were measured; (ii) in the second meeting we collected self-reported measures of psychological and social wellbeing; and (iii) in the third meeting a personal trainer performed a physical assessment. The latter allowed for personalised tailoring of exercise type and progression levels (in terms of duration, use of weights, and number of repetitions), and for personalisation of the starting level of each participant. Measurements were performed by the Coach and the sociologist of the local senior citizen organization. Both were not aware of the group allocation at the time of the measurements.

The study took place in Trento, Italy, over a period of 10 weeks from October to December 2014. The duration and size of this study is similar to previous technology-supported interventions that have seen significant results in adherence and improvements physical measures (e.g., *Silveira et al., 2013b*). The first week was devoted to technical deployment and application testing, followed by 8 weeks of training and 1 week of post-training measurements. The training program was supervised by a training coach, who could intervene to advance trainees in the exercise program, and to provide technical

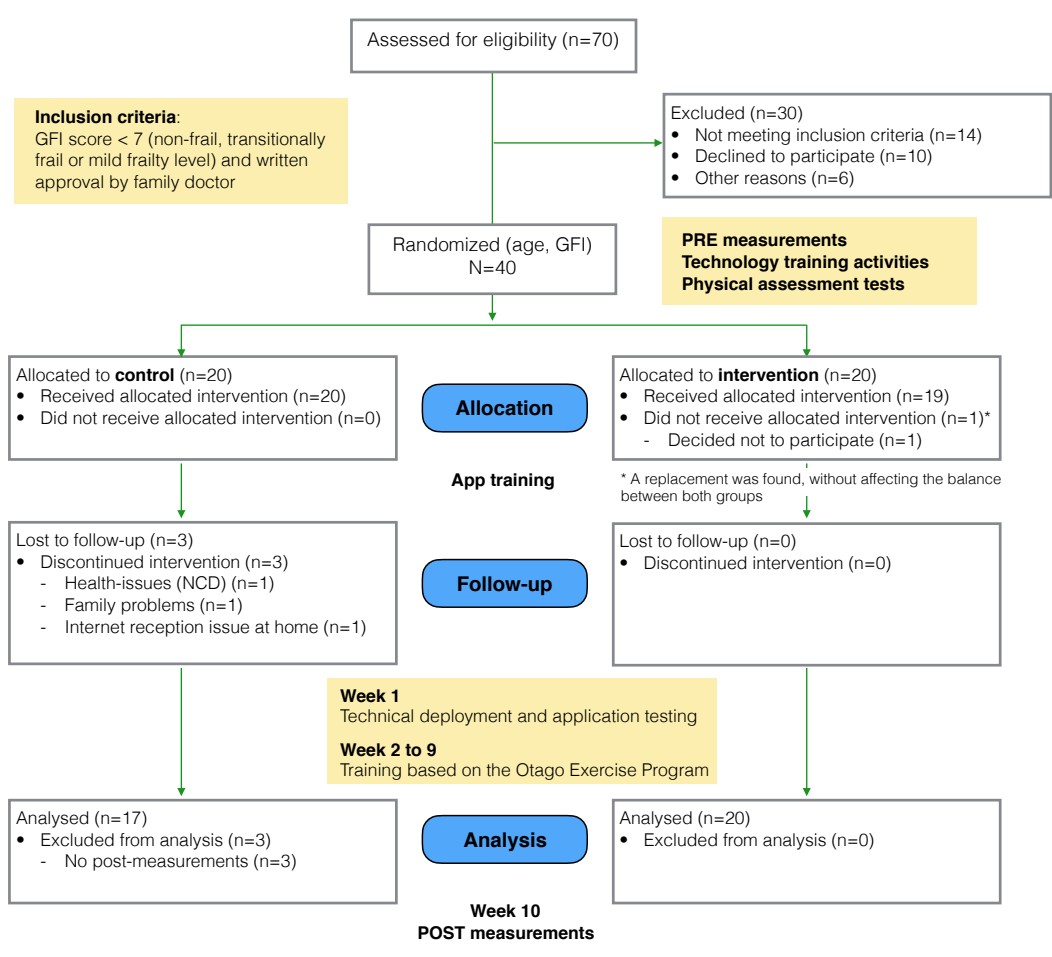

**Figure 2** Study flow diagram.

support upon request. The same level of technical support was made available to both groups.

The study received ethical approval from the CREATE-NET Ethics Committee on ICT Research Involving Human Beings (Application N. 2014-001). As the study follows a framework for the design of complex interventions in healthcare (*Campbell et al., 2000*), at this stage it is considered as a pilot study.

## Participants

We considered eligible for the study participants aged 65 or older, independent-living, self-sufficient and with a non-frail, transitionally frail or a mild frailty level. The latter was measured using the Groningen Frailty Indicator (GFI; *Steverink et al., 2001*), considering eligible those participants scoring lower than 7 in the scale from zero (not frail) to fifteen (very frail). Participants wearing pacemakers were not considered eligible, since the study required the use of a mobility sensor, as well as participants not able to undergo the exercise program according to their family doctor.

**Table 1 Summary statistics for experimental and control groups.**

| | Control ($N = 20$) | Experimental ($N = 20$) | *p*-value |
|---|---|---|---|
| Age, mean (SD) | 71.5 (6.809) | 70.3 (4.485) | 0.515[a] |
| Females, % | 75% | 70% | 1.000[b] |
| GFI, mean (SD) | 2.45 (1.638) | 3.050 (1.849) | 0.284[a] |
| RAPA, mean (SD) | 5.45 (1.317) | 5.5 (1.235) | 0.902[a] |
| Self-reported measures, after allocation[*] | | | |
| PACES, mean (SD) | 3.837 (0.584) | 4.014 (0.782) | 0.444[a] |
| Loneliness score (SD) | 6.312 (2.387) | 5.8 (2.858) | 0.562[a] |
| Subjective wellbeing score (SD) | 7.25 (2.72) | 7.5 (3.517) | 0.811[a] |
| Physical assessment, after allocation[*] | | | |
| Starting level, mean (SD) | 2.375 (0.885) | 3.400 (0.821) | **0.003[a]** |
| Leg muscle strength score (SD) | 12.312 (3.894) | 15.4 (4.235) | **0.029[a]** |
| Gait speed score (SD) | 0.676 (0.156) | 0.809 (0.16) | **0.017[a]** |

**Notes.**
[a] Differences computed using independent samples *t*-test.
[b] Differences computed using Pearson Chi squared test.
[*] Control group was reduced to 17 participants after dropouts.

We recruited participants through members of local associations (*Ada* and *Auser*) that promote initiatives for elderly persons in Trento, Italy. We sent invitations to the 70 persons that visited the associations more recently. Out of these, 10 persons declined the invitation, 6 were excluded because they lived in an area without 3G or LTE coverage, and 14 did not meet the frailty criteria. In the end, a total of 40 participants between 65 and 87 years old were recruited for the study (29 females and 11 males, mean age = 71, s.d. = 5.7). All participants obtained a formal written approval by their family doctor to allow them to participate in the study. Both doctors and participants received a written outline and explanation of the study and signed the consent before participating.

From the initial group of participants, 4 older adults withdrew at different times during the course of the study due to unpredictable health or family problems. One participant was substituted because the withdrawal occurred before the beginning of the study, while the others could not be replaced since they withdrew during the course of the study. For this reason, the results are based on the data from 37 participants (27 females and 9 males, mean age = 71.2, s.d. = 5.8, between 65 and 87 years old).

Participants were not told to which group they were assigned or that a different version of the application was being tested.

In relation to the technology, less than 20% of the participants had ever used a tablet before, and less than 10% used it regularly. Thus, and as mentioned in the previous subsection, all participants were provided introductory courses on how to use the tablet.

In Table 1 we summarise the initial measures for both groups. A *t*-test for independent groups shows no statistical difference between control and experimental groups in terms of the initial measures, except for the ones related to the physical assessment. This issue is addressed later in the analysis.

### Intervention: activity program

The exercise program implemented in this study was developed on the basis of the OTAGO Exercise Program (*Gardner et al., 2001*), and was adapted by a professional personal trainer in order to fit the original program into 10 levels of increasing difficulty (Table S1).

The OTAGO Exercise Program is used worldwide and is one of the most tested fall prevention programs (with four randomized controlled trials and one controlled multi-center trial (*Stevens, 2010*)). The program includes muscle strengthening and balance-retraining exercises of increasing levels in terms of duration and repetitions. The duration of the exercise sessions ranged from 30 to 40 min, with longer sessions in the higher levels.

Participants from both the social and control group were assigned an initial level by the Coach based on the pre-test analysis. As a minimum requirement, participants were then asked to participate in two exercise sessions per week. Participants of both groups were able to progress in the exercise program every week, via an automatic level-up suggestion by the system. If participants agreed to level-up, the personal trainer would verify the attendance and completeness of exercises before enabling the following level.

## TEST PROCEDURES AND OUTCOME MEASURES

### Primary outcomes

#### Participation measures

**Attrition**. The attrition rate was used to measure the proportion of participants lost at the end of the study.

**Adherence**. The adherence was used to measure the conformity of the participants with the exercise program. For each participant, two measures were considered. The first is related to *persistence* throughout the eight weeks of the exercise program, and it was computed considering the ratio between the number of participations in exercise sessions by a participant and the number of the exercise sessions planned in the program. Participation was measured by logging the attendance to the scheduled training sessions in the virtual classroom (considering also partial participation, where participants skipped exercises). The second measure is related to the level of *completeness* of the exercise sessions. It was calculated considering, for each session, the percentage of exercises videos that the user actually followed (watched)—excluding the time of preparation and skipped exercises—with respect to the total duration of the exercises planned for the session.

#### Physical assessment exercises

Specific assessment exercises, developed and validated within the Otago Exercise Program (*Campbell et al., 1997*; *Campbell & Robertson, 2003*), were used to measure participants' leg muscle strength and walking ability at the beginning and at the end of the study, in a face-to-face session with each participant. In particular, the assessment exercises were:

- **30 s Chair Stand** test (*Jones, Rikli & Beam, 1999*): the purpose of this test is to evaluate leg strength and endurance. From seated position, the participant rises to a full standing position and then sit back down again for 30 s. The outcome measure is the number of times the participant comes to a full standing position in 30 s.

- **Timed Up & Go** test (*Podsiadlo & Richardson, 1991*; *Rossiter-Fornoff et al., 1995*): the purpose of this test is to assess older adult mobility. From the seated position, the participant stands up from the chair, walks for 3 m at his/her normal pace, then turns, walks back to the chair and sits back down again. The outcome measure is the number of seconds to complete the test.

## Secondary outcomes
### Psychological dimensions

We investigated changes in psychological dimensions related to different aspects of physical activity and wellbeing, by collecting participants' feedback before and after the study on the following measures:

**Enjoyment of physical activity**. Past literature has shown that intrinsically motivated people tend to engage in physical activity for personal improvement and because they enjoy it (*Deci & Ryan, 1985*; *Pelletier et al., 1995*). Enjoyment of physical activity is believed to develop positive attitudes toward exercise, enhance intrinsic motivation, and consequently foster long-lasting adherence to physical activity (*Ryan et al., 1997*; *Wankel, 1993*). In order to measure participants' enjoyment of physical activity at the beginning and at the end of the study, we used the Physical Activity Enjoyment Scale (PACES; *Kendzierski & DeCarlo, 1991*), which has been validated in several studies, including one with an Italian sample (*Carraro, Young & Robazza, 2008*). The scale includes 16 items scored on a 5-point Likert scale with the range from 1 (disagree a lot) to 5 (agree a lot). Total enjoyment scores range from 16 to 80 (maximum enjoyment).

**Subjective wellbeing**. In order to investigate the effectiveness of the application and training in improving subjective wellbeing, we collected participants' feedback before and after the training period using the Wellbeing scale of the Multidimensional Personality Questionnaire (MPQ; *Tellegen & Waller, 2008*). This scale was developed to assess positive emotional tendencies as distinct from the absence of a negative emotional disposition. Wellbeing represents individual dispositions to experience positive emotions, and is an important marker of the higher order Positive Emotionality dimension. The scale includes 12 items requiring a true / false response, with the total scoring ranging from 0 to 12. People who score higher in this scale tend to describe themselves as cheerful, optimistic, hopeful, having interesting experiences and engaging in enjoyable activities (*Tellegen & Waller, 2008*, p. 274).

### Social wellbeing

The effects of the technology-based intervention on the social wellbeing was assessed on the basis of participants' feedback, collected at the beginning and at the end of the study using a measure of loneliness.

**Loneliness** is an aspect that negatively affects social wellbeing. Shifts in the social environment, and in particular loneliness, are believed to be an important aspect in the life of aging people (see for example *Hughes et al., 2004*; *Liu & Rook, 2013*). Loneliness involves individual perception of social isolation and feelings of not belonging and being disconnected, and is a central aspect of a group of socio-emotional states including, among others, self-esteem, optimism, anxiety, anger and social support. To measure loneliness,

we used a shorter version of the R-UCLA Loneliness Scale (revised version of the UCLA Loneliness Scale developed by University of California, Los Angeles) (*Russell, Peplau & Cutrona, 1980*) developed by *Hughes et al. (2004)*. The scale used includes 3 items scored on a 5-point Likert scale, with the total score ranging from 3 to 15, and higher scores indicating higher levels of loneliness.

## STATISTICAL ANALYSES

**Program Adherence**. We analyze *persistence* with an analysis of variance (ANOVA) with group (social vs. control) and initial scores of *gait speed*, *leg muscle strength* and PACES (enjoyment of physical activity) as between-subject factors. *Gait speed*, and *leg muscle strength*, and PACES scores are grouped into three equally distributed intervals: Low, Medium, and High. In this ANOVA, we take into account only the interactions between group and the other three factors.

**Gait speed and leg muscle strength**. We analyzed Gait speed (in terms of *Timed Up & Go* test score) with a repeated-measures analysis of covariance (ANCOVA) with group (control vs. social) as between-subject factor, time (pre- vs. post-measurement) as within-subject factor, and persistence as covariate. In the ANCOVA, we take into account the interaction between group and time. We perform the same analysis for muscle strength (in terms of *30 s Chair Stand* test score).

**Enjoyment of physical activity**. We perform a repeated-measures of covariance (ANCOVA) to determine a statistically significant difference in the *PACES scores* between control and social group, using pre- and post- measurement points as within-subject factor, and persistence as covariate. We compute the main effect for time, and the interaction between time and group.

**Subjective wellbeing**. We used the median value to dichotomise this variable into "Low" (respondents with less than or equal to median subjective wellbeing score) and "High" wellbeing (respondents with more than median score). We analyze the subjective wellbeing score by means of a logistic regression with group (control vs. social), time (pre- vs. post-measurement), and persistence as factors. The model includes also the interaction effect of time with both group and persistence. In the logistic regression, "High" is used as reference level of the dependent variable.

**Loneliness**. We used the median value (i.e., 5) to dichotomise this variable into "Low" (respondents with less than or equal to median loneliness score) and "High" loneliness (respondents with more than median loneliness score). This binary variable is analyzed by means of a logistic regression with group (control vs. social), time (pre- vs. post-measurement), and persistence as factors. The model includes also the interaction effect of time with both group and persistence. In the logistic regression, "High" is used as reference level of the dependent variable.

In order to interpret significant interactions, we run a post-hoc *t*-test corrected with Bonferroni.

We perform the analyses using the open source statistical software R (*R Core Team, 2013*), using the ggplot2 (*Wickham, 2009*) and ggnet (*Schloerke et al., 2016*) packages for

plotting the results. We consider as extreme values the data points in the 1.5 interquartile ranges (IQRs) below the first quartile or above the third quartile. Results excluding extreme values are reported when the normality assumption of the test is not met.

## RESULTS

### Application usage

Before reporting on the main outcome measures, we summarise the results from *Baez et al. (2016)*, which reported on how participants of the intervention—also studied in this paper—perceived and used the trainee application. We also report on online social interactions, to put the social wellbeing outcome in context.

#### Participants' perception

To collect feedback on the participant's perception of the tool, we used a questionnaire (https://goo.gl/zl7daL) assessing the most stimulating aspect of the overall experience and the usefulness of the various features.

*Most stimulating aspects.* A manual classification using a emerging coding scheme was performed to identify main themes in participants' open-ended responses.

As *stimulating aspects*, two main themes emerged in the **social group**: training with others and the possibility of following the training program. Other participants expressed the possibility of messaging or challenging themselves. In the **control group** the possibility of training from home was also a main theme along with the personal satisfaction of doing the exercises. Interestingly, in this group one person reported that "The experience was interesting, though it was a pity that the Coach was in the video and not actually present".

As *negative aspects*, the dominant issue was the intermittent interruptions in the Internet service that occurred at some point during the study, and which affected both groups equally.

*Usefulness of features.* Participants of the **social group** continued to use all features of the application throughout the study, although with different perceived usefulness. As reported in *Baez et al. (2016)*, the features that are instrumental to the training were naturally experienced by most of the trainees, and this includes exercising in the classroom, checking out the schedule, and more importantly, training with the company of others. Together these features were highly valued. Persuasion features were also among the most experienced and valued. This includes, following the progress and visualizing their own progress in the garden and, still very positive but to a lesser extent, inviting others to join a training session

Interestingly, social interaction features received mixed results. While group chat was used widely, personal messaging was heavily used to interact with the Coach but less with other participants.

The results above show not only that participants of the social group **did feel as if they were training with others** but also that they considered it as one of the most stimulating aspects. This supports the results reported in *Far et al. (2015)*, where it was observed that

participants in an online group setting resulted in a significantly higher number of joint sessions (training together as opposed to training alone) compared to the control group.

### Online social interactions

We briefly summarise the usage of the online social interaction features that were available in the social group: private messages and bulletin board (public messages). The usage was analyzed by looking at the database of messages exchanged among participants.

Private messages were preferred over the public messages, accounting for 75% of all the messages exchanged (544 messages). The most active user was the Coach, who had to contact the trainees to check on their progress on a weekly basis, followed by the Technician. We illustrate the interactions between all the participants in Fig. S1 .

A qualitative content analysis of the online interactions was presented in *Baez et al. (2016)*, where we reported the distinctive use of private and public messages. We developed a coding scheme based on relevant literature about online behaviour and communities, developing a final coding scheme composed of 5 top- and 12 sub-categories. Then two independent researchers coded the messages (Cohen's kappa: public messages .85 and .84 for top and sub categories; private messages .87 and .85 for top and sub categories correspondingly).

We highlight some observations to put in context the results in this section:

- In private messages the Coach took an active role, dominating the discussions around physical activity and, in particular, by offering support. Trainees instead lead discussions focused on community building, and entertainment (e.g., sharing jokes). When discussing about physical activity, trainees focused on reporting their personal experience with the exercises. Discussions about the application were shared mostly with the technician and were related to issues with the application.
- In the bulletin board, the Coach was much less active, limiting his participation to congratulating users after each week of training. Trainees participated more actively in community building, and in discussions about physical activity where they also provided support and encouragement to their peers. Positive comments about the application were interestingly largely more predominant in public.

Participants of the control group enjoyed the same type of support, although via phone calls, diaries and transcripts of the interactions with the Staff were not available, limiting the comparison of effects at the group level and the detailed analysis of interactions to the Social group.

## Program adherence

The results initially described in *Far et al. (2015)*, provided interesting insights on the effect of social features on participants' adherence. In this work, we extend on the previous analysis to report on the effect of participants' physical abilities and attitudes towards exercising on their adherence to the training.

### Attrition

The intervention resulted in a 7.5% attrition rate (corresponding to 3 participants), measured in terms of the proportion of participants lost at the end of the study. Reasons

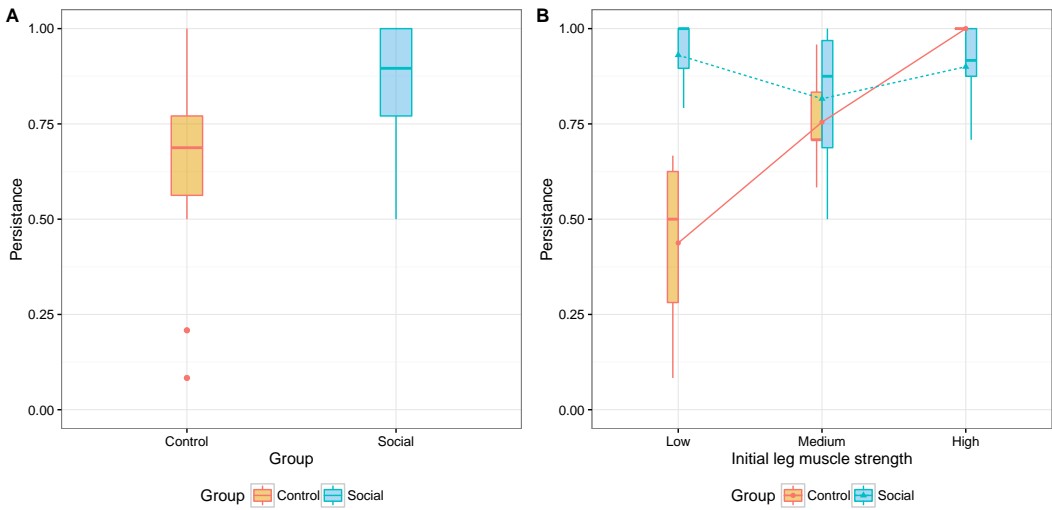

**Figure 3** Interaction plots for persistence and (A) initial measures of leg muscle strength and (B) physical activity enjoyment.

behind the withdrawal of these participants were related to unexpected health and family problems or, in one case, because of Internet connection issues which could not be solved.

### Persistence

The overall persistence rate in the two groups was of 76% (SD = 22.6%), when considering the total number of sessions available (Fig. 3A). In the social group the persistence rate was 85%, while in the control group it was 64%. A between-subjects analysis of variance was performed to compare the persistence of both groups while controlling for the initial scores in the measures of gait speed and leg muscle strength (physical measures), and the enjoyment of physical activity. The independent variables were grouped in three equally distributed intervals (Low, Medium, High). The analysis showed a significant interaction between group and the initial measures of leg muscle strength ($F(2, 23) = 5.966$, $p = .008$, partial eta squared = .342), but no significant interaction with gait speed ($F(1, 23) = 3.42$, $p = .08$, partial eta squared = .13) nor enjoyment of physical activity ($F(2, 23) = 1.93$, $p = .17$, partial eta squared = .144). In Fig. 3B we show the relevant interaction plot. The same analysis, exploring the effect of the initial loneliness score as independent variable (with and without controlling for the other variables) reveal no significant interaction with group.

The interaction between group and leg muscle strength shows a higher persistence in participants of the control group that scored higher in the initial leg muscle strength test. For the social group however, there is no significant difference in the persistence of participants based on their initial leg muscle strength score. This suggests that, when training individually, fitter participants tend to adhere more to the training, but when social elements are in place these differences in fitness no longer determine adherence.

There was also a significant main effect for group ($F(1, 23) = 13.151$, $p = .001$, partial eta squared= .364), with the social group showing a higher persistence rate ($M = 85.4\%$, SD = 16.1%) compared to the control group ($M = 64.2\%$, SD = 24.1%). Considering

the minimum number of sessions per week as recommended by the Coach (2 sessions per week), it results in a higher level of persistence for both groups: social group ($M = 97.5\%$, $SD = 6.8\%$) and control group ($M = 85\%$, $SD = 25.9\%$).

The lower variability of persistence in the social group can be explained by the social features (normative influence, social facilitation, social learning) that might have motivated users to comply with the community norm. In the control group, the higher variability suggests a stronger effect of individual differences due to the lack of social awareness.

### Completeness

The overall completeness rate in the two groups was 90.32% ($SD = 17.4\%$), suggesting that participants tended to complete the working sessions once they started. The completeness rate in the social group ($M = 91.75\%$, $SD = 12.46\%$) was slightly higher compared to that of the control group ($M = 88.63\%$, $SD = 22.24\%$), although not statistically significant. As in the previous measure, a between-subjects analysis of variance was performed to compare the completeness of both groups while controlling for the initial scores in the measures of gait speed and leg muscle strength, enjoyment of physical activity and loneliness. No interaction was found between group and the initial scores, but a main effect for initial leg muscle strength ($F(2, 23) = 5.075$, $p = .015$, partial eta squared $= .306$). We attribute this effect to the duration of the video exercises that were assigned to fitter individuals (e.g., required more repetitions), which were longer for higher levels of intensity.

The higher level of completeness and lower variability in the social group can be explained by the presence of self-monitoring tools (e.g., positive and negative reinforcement) and social facilitation (exercising with others), which were lacking in the training sessions of the control group participants.

## Muscle strength and gait speed

Two types of assessment exercises, developed and validated within the OTAGO Exercise Program (*Campbell et al., 1997*; *Campbell & Robertson, 2003*), were used to measure participants' leg muscle strength and walking ability at the beginning and at the end of the study.

### Leg muscle strength

A repeated-measures of covariance (ANCOVA) was performed to determine a statistically significant difference in the *30 s Chair Stand scores* between control and social group, using pre- and post- measurement points as repeated measures variable, and persistence as covariate (Fig. 4A). The analysis showed a significant main effect for time ($F(1, 29) = 37.803$, $p < .001$, partial eta squared $= .566$) but no interaction between group and time ($F(1, 29) = 0.704$, $p = .404$, partial eta squared $= .024$). This suggests that, while there is an overall improvement, time did not have a substantially different effect on the performance of the two groups nor on participants with different levels of adherence. We attribute this effect to (i) high adherence with the minimal recommendations by the Coach and (ii) the duration of the pilot that might not have been enough to see statistically significant results.
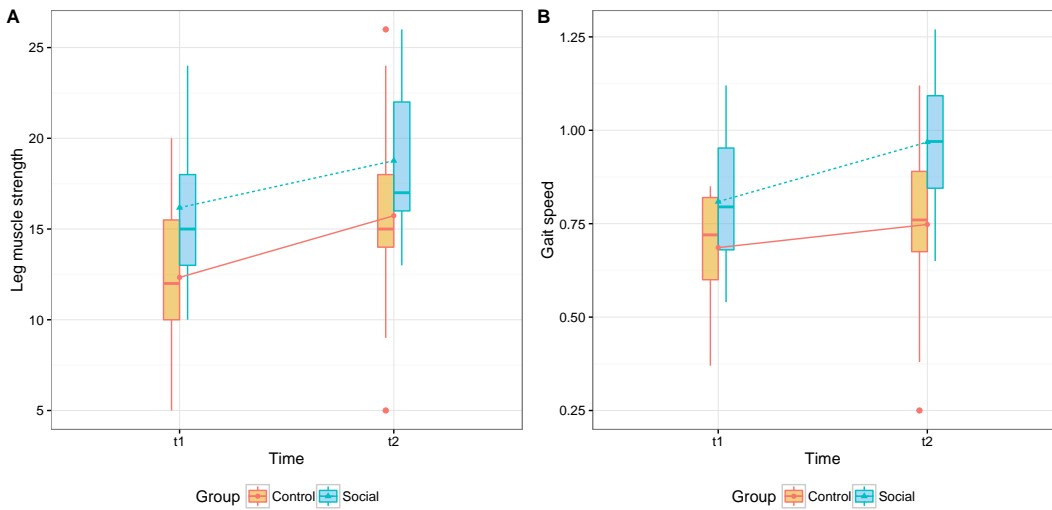

**Figure 4** Physical outcomes before and after eight weeks of training (A) participants' scores in the 30 s chair stand, measuring leg muscle strength, (B) participants' gait speed in the Timed Up & Go test, measuring walking ability.

The analysis also revealed a main effect of group ($F(1, 29) = 6.809$, $p = .014$, partial eta squared $= .19$) and persistence ($F(1, 29) = 10.233$, $p = .003$, partial eta squared $= .261$). We should note that despite randomization and the non-significant difference between the self-reported physical activity in the two groups (measured with the Rapid Assessment of Physical Activity Questionnaire by *Topolski et al. (2006)*), participants in the Social group performed better than participants in the Control group in the 30 s Chair Stand pre-test.

### Gait speed

A repeated-measures of covariance (ANCOVA) was performed to determine a statistically significant difference in the *Timed Up & Go scores* between control and social group, using pre- and post- measurement points as repeated measures variable, and persistence as covariate (Fig. 4B).

The analysis showed a significant main effect for time ($F(1, 31) = 11.952$, $p = .002$, partial eta squared$= .278$) but no interaction between group and time. The results bear similarities with the ones of leg muscle strength test, suggesting overall improvement in gait speed after the training program, but not a significant different effect on the two groups nor on participants with different levels of adherence. The analysis also revealed a main effect of group ($F(1, 31) = 11.789$, $p = .002$, partial eta squared $= .276$) and persistence ($F(1, 31) = 5.086$, $p = .031$, partial eta squared $= .141$). As in the previous measure, we attribute the effect to the overall high adherence and to the duration of the pilot study.

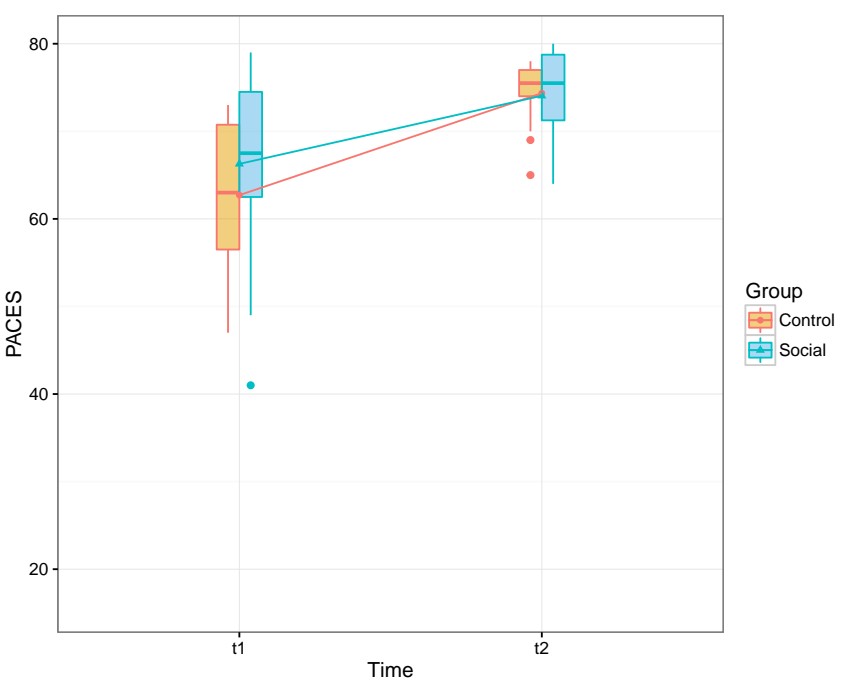

**Figure 5** Participants' mean scores in the physical activity enjoyment scale before and after the eight-week period of the exercise program, excluding extreme values from the original dataset.

## Psychological dimensions
### Enjoyment of physical activity

We used the Physical Activity Enjoyment Scale (PACES; *Carraro, Young & Robazza, 2008*; *Kendzierski & DeCarlo, 1991*) to measure participants' enjoyment of physical activity at the beginning and at the end of the study (Fig. 5).

A repeated-measures of covariance (ANCOVA) was performed to determine a statistically significant difference in the *PACES scores* between control and social group, using pre- and post- measurement points as repeated measures variable, and persistence as covariate. The analysis showed a main effect for time ($F(1, 33) = 16.998$, $p < .001$, partial eta squared $= .278$) but no interaction between group and time. An analysis excluding extreme values resulted in comparable results, with a main effect for time ($F(1, 31) = 23.297$, $p < .001$, partial eta squared $= .429$). These results show that participants enjoyed more engaging in physical activity at the end of the study regardless of the group and their level of adherence.

The effect of the initial level of physical ability on the enjoyment was also explored (muscle strength and gait speed as covariates in the model) but no effects were found.

### Subjective wellbeing

Subjective wellbeing was measured by means of MPQ (*Tellegen & Waller, 2008*) before and after the study (Fig. 6). In the logistic regression performed on the subjective wellbeing score, only time was a significant factor ($B = -1.050$, $OR = 0.350$, 95% CI [0.129–0.909],

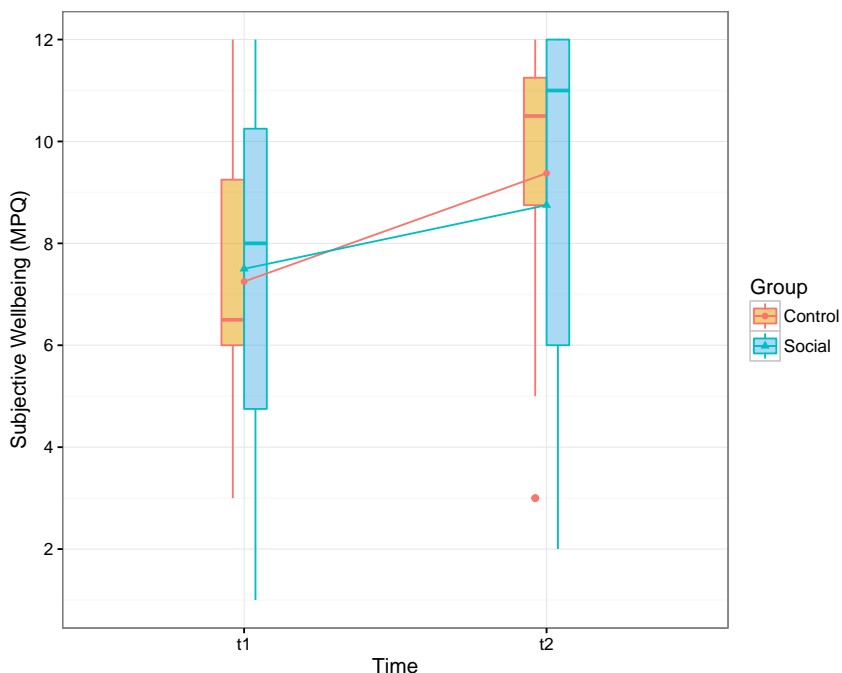

**Figure 6** **Participants' scores in the wellbeing scale of the multidimensional personality questionnaire before and after the training period (range, 1–12).**

$p = .034$). This suggest that subjective wellbeing improved for participants of both groups, regardless of the version of the app.

## Social wellbeing
### Loneliness

In the logistic regression performed on the loneliness score, only the factor time was significant in improving (reducing) loneliness ($B = 1.121$, $OR = 3.068$, 95% CI [1.177– 8.380], $p = .024$). To investigate the extent of the improvement according to the initial loneliness score (t1) we also tested the interaction between that initial score (grouped in three equally distributed intervals: Low, Medium, High) and time. Although the interaction is not statistically significant we can see a stronger trend for higher initial levels of loneliness. In Fig. 7 we illustrate the loneliness scores before and after the training.

These results suggest that, overall, the perception of loneliness significantly decreased after the training, regardless of the group and adherence to the training. We attribute this effect to the attention given by the Coach to both groups.

To investigate if the use of social features predicts the improvement in the loneliness score for the social group, we performed a correlation test (with Spearman method) using the number of private and public messages as predictors (Fig. 8). For private messages we took message received, as the exchanges were nearly symmetrical. The results show that number of messages received is significantly correlated with the improvement in the loneliness scores (rho $= -.635$, $p = .003$). Public messages, on the other hand, were not correlated (rho $= -.244$, $p = .314$).

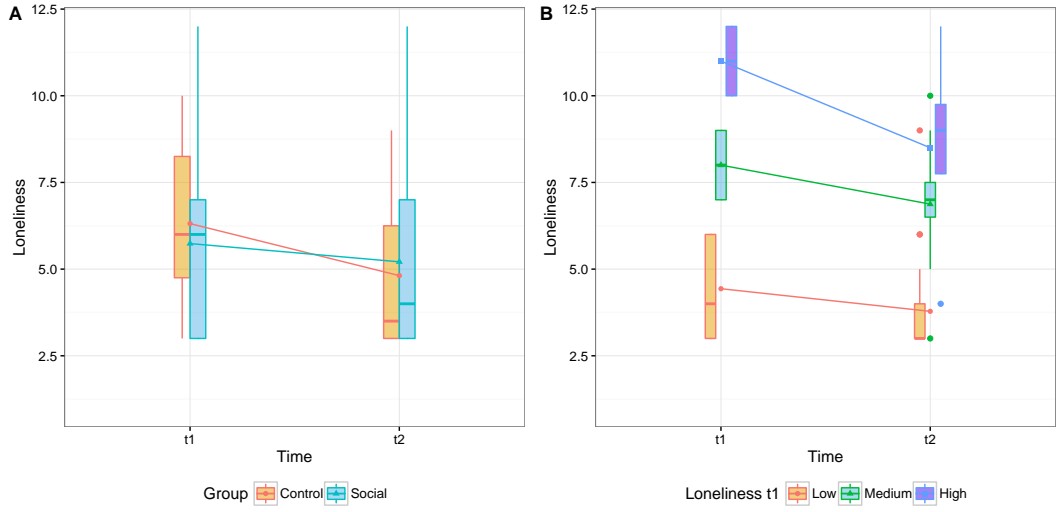

**Figure 7** Participants' mean scores in the abbreviated form of the R-UCLA Loneliness Scale before and after the eight-week period of the exercise program.

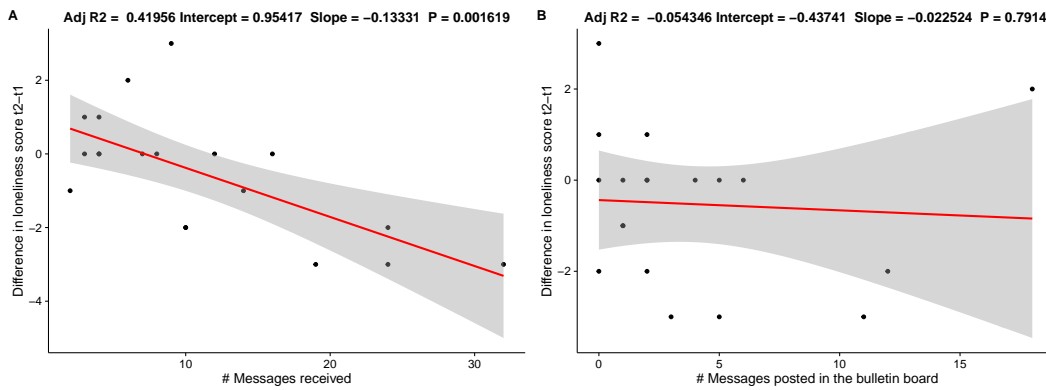

**Figure 8** Regression lines for (A) number of messages received and (B) number of public messages posted.

# DISCUSSION

## Main findings

### *In virtual group exercising, adherence by persons with low starting levels of physical skills reaches the same level as that of more fit participants*

Training adherence outcomes show that the initial level of skill had no significant influence on the adherence of participants of the Social group, while in the Control group we observed that fitter participants tended to adhere more to the training. These results suggest that

(i) the online group-exercising could potentially overcome a major issue reported in the literature (*De Groot & Fagerström, 2011*) in terms of negative effect of group-exercising in the motivation of heterogeneous groups, and

(ii) it helped in reducing the effect of the initial level of skill in the motivation of participants, with trainees complying to the group norm.
We should notice that despite the small sample we have observed a large effect size (partial eta squared = .327). This pilot thus motivates further studies into the effect of baseline measures on adherence.

In addition, we have seen evidence of the preference of older adults for group exercising. This comes from the debriefing of participants, and after observing the participation of trainees in the Social group, who were able to choose whether to train alone or in company, and largely participated of training sessions in company (after controlling for casual meet-ups) (*Far et al., 2015*). This is in line with existing literature pointing to the importance of social features in fitness applications (*Far et al., 2016*).

These results contribute to the literature studying the effects of group-based and home-based training, which on short-term settings have not provided conclusive results (*Van Der Bij, Laurant & Wensing, 2002*; *Freene et al., 2013*). However, the hybrid nature of the proposed intervention requires further studies, to compare its effectiveness to that of traditional interventions.

### Virtual group-exercising enables tailored home-based intervention with positive physical outcomes and increased persistence

In terms of physical outcomes, at the end of the eight-week program, both Social and Control groups showed significant improvement in gait speed and leg muscle strength, given the high levels of adherence in both groups. It is important to note that by providing the feeling of *training together* to an otherwise *tailored* exercise program, participants of the Social Group observed the benefits of performing exercises that were tailored to their individual abilities (traditionally part of an individual training program) while enjoying the extra motivation of the social context (as indicated by the greater *persistence* of the social group).

These results are in line with previous literature pointing to the equivalent health-related outcomes of traditional group- and home-based training (*Freene et al., 2013*), though in this setting we have achieved these results with a heterogeneous group. However, further studies are required in order to observe these effects in long-term settings.

### Positive effects on enjoyment and subjective wellbeing at the end of the training program (regardless of the control or social condition)

The psychological measures have shown a significant improvement after the eight-week training program. Participants from both groups showed a significant increase in the *enjoyment of physical activity*, as measured by the Physical Activity Enjoyment Scale (PACES; *Carraro, Young & Robazza, 2008*; *Kendzierski & DeCarlo, 1991*), and in *subjective wellbeing*, as measured by the MPQ (*Tellegen & Waller, 2008*). This supports the literature associating regular physical activity with positive outcomes in health and wellbeing in later age (*Thibaud et al., 2012*; *Stuart et al., 2008*; *Landi et al., 2010*). However, the difference in the adherence observed between both groups did not account for a statistical difference in psychological outcomes. Further analysis is required to understand whether these measures are not affected by the exercise settings or differences in the software features.

### Decrease in loneliness in both groups, attributed to contacts with the Coach (and between participants)

We also observe that, although social features played a part in motivating participation to training sessions in the Social group (*Far et al., 2015*), there was no significant difference in terms of loneliness scores with respect to the Control group after the eight weeks of training. A plausible explanation for the improvement in the Control group is the frequent contact with participants via telephone. This is common practice in clinical evaluation studies involving older adults, where social visits to the control group are performed to mirror the time and attention provided to the treatment group, and even suggested by several authors (for example, *Hogan et al., 2001*; *Michael et al., 2010*; *Tinetti et al., 1994*). In our study, each participant in the control group was contacted by phone, rather than via the messaging features of the application, and it is possible that the effect of this contact was strong enough to produce a significant decrease in participants' perception of loneliness. Indeed, previous work relying on the R-UCLA loneliness scale, and with similar intervention periods (6–15 weeks), have achieved significant reductions of loneliness but have also included the physical presence of educators, trainers, and peers during the intervention (*Shapira, Barak & Gal 2007*; *Fukui et al., 2003*).

Thus, we found noteworthy the significant decrease in the level of loneliness in the Social group despite the use of remote interactions and in such a short period. What is more, the online interactions in the form of private messages were found to predict the decrease in loneliness.

## Limitations

The complexity of the study setting resulted in limitations that are acknowledged in the following:

**Different tools for support**. The interactions of the Coach with the participants were part of the study protocol and designed to give the same type of support. However, while in the Social group the communication was carried on within the app via messaging features, in the Control group it was done via phone. This was done so given the absence of social features in the version of the app used by the Control group. This difference in the communication—i.e., using a more direct channel in the Control group—might have introduced a potential bias in the social wellbeing outcomes in favour of the Control group.

**Sample size and gender imbalance**. Random variability, probably due to the small sample size, might have influenced the initial difference between groups in some of the measures. Although the comparisons reported in the paper were done in terms of relative improvements and not strict comparisons, this should be noted as a potential bias.

The gender imbalance, resulting in a skewed female to male ratio, should also be noted as a potential limitation. Previous studies, however, provide evidence in favour of the generalization of our results, noting that male and female react equally to sport, despite differences in initial motives for participation (*Koivula, 1999*; *Ryan et al., 1997*).

**Duration of the training**. The pilot intervention ran for a period of two months, which was significant for observing differences in adherence and even some effects in physical measures. However, the OTAGO program this intervention is based on, relied on a

longer duration (four months). Thus, this constitutes a limitation of our study, for more meaningful comparisons and outcomes might be observed in longer periods.

**No cognitive measures at baseline**. While the usability of the application was among the pre- and post- measures, we did not include any standard instrument for measuring cognitive abilities of individuals, as to relate the usability (and other outcomes) to the cognitive abilities of the participants. However, as reported in this paper, participants were independent-living older adults that did not show any issues during the tablet and application training sessions.

## CONCLUSION

In this paper we introduced a technology-based physical intervention to enable older adults with different abilities, and indeed *in spite* of their different abilities, to engage in group exercises from home while keeping these differences invisible to the group. We focused particularly on the feasibility of delivering a tailored exercise program while keeping the feeling of training in a group, and measuring the effects of such an intervention on the adherence of trainees of different abilities, and on the physical outcomes. In addition, we explored the effects of the intervention on psychological and social wellbeing outcomes.

The results indicate that technology-supported online group exercising which conceals individual differences in physical skills is effective in motivating and enabling individuals who are less fit to train as much as fitter individuals. This not only indicates the feasibility of training together *despite* differences in physical skills but also suggests that online exercise can reduce the effect of skills on adherence in a social context. Longer term interventions with more participants are instead recommended to assess impacts on wellbeing and behavior change.

## ACKNOWLEDGEMENTS

We thank Elena Isolan for her collaborative efforts, Associazione per i Diritti degli Anziani (A.D.A), Associazione per l'invecchiamento attivo (Auser), and Smart CROWDS Trento and their staff for their help and support during the study.

### Funding

This work was performed in collaboration with Tomsk Polytechnic University within the project in Evaluation and enhancement of social, economic and emotional wellbeing of older adults under the Agreement No.14.Z50.31.0029. The project was also funded by the European Institute of Technology (EIT Digital) under the grant "Personal Fitness Club". The grants paid for the research work on the definition of the study protocol and execution of the study. The funders had no role in study design, data collection and analysis, decision to publish, or preparation of the manuscript.

## Grant Disclosures

The following grant information was disclosed by the authors:

Tomsk Polytechnic University.

European Institute of Technology (EIT Digital).

## Competing Interests

Marcos Baez, Iman Khaghani Far, Francisco Ibarra, Fabio Casati are researchers at their corresponding institutions and are also associated with Gymcentral, a training platform maintained by the University of Trento and freely available to researchers at http://gymcentral.net; however, they do not receive compensation from Gymcentral nor did Gymcentral have any role in their research or publications.

## Author Contributions

- Marcos Baez conceived and designed the experiments, performed the experiments, analyzed the data, contributed reagents/materials/analysis tools, wrote the paper, prepared figures and/or tables, reviewed drafts of the paper.
- Iman Khaghani Far conceived and designed the experiments, performed the experiments, contributed reagents/materials/analysis tools.
- Francisco Ibarra performed the experiments, contributed reagents/materials/analysis tools, wrote the paper, reviewed drafts of the paper.
- Michela Ferron conceived and designed the experiments, performed the experiments, analyzed the data, wrote the paper.
- Daniele Didino analyzed the data, wrote the paper, reviewed drafts of the paper.
- Fabio Casati conceived and designed the experiments, performed the experiments, contributed reagents/materials/analysis tools, wrote the paper, reviewed drafts of the paper.

## Human Ethics

The following information was supplied relating to ethical approvals (i.e., approving body and any reference numbers):

The study received ethical approval from the CREATE-NET Ethics Committee on ICT Research Involving Human Beings (Application N. 2014-001).

## Data Availability

The raw data has been supplied as a Supplementary File.

## Supplemental Information

Supplemental information for this article can be found online at http://dx.doi.org/10.7717/peerj.3150#supplemental-information.

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
