# Peer review of "Effects of online group exercises for older adults on physical, psychological and social wellbeing: a randomized pilot trial"

_PeerJ, doi:10.7717/peerj.3150_

## Round 0.1 · original submission · Major Revisions

Although the reviewers noted that manuscript has improved significantly, they raised questions about the low sample size of the study. The authors should make sure that all results are interpreted consistent with the limited sample and clarify the rational for reporting a pilot trial.

·

Basic reporting

“The OTAGO Exercise Program is used worldwide and is one of the most tested fall prevention programs by the Centers for Disease Control and Prevention (with four randomised controlled trials and one controlled multi-center trial [Stevens, J. A., 2010]).”  Please re-phrase as the current sentence indicates that this program is evaluated by CDC.

Please spell out “R-UCLA” when first appears in the text.

Experimental design

Several issues should still be cleared to increase transparency, quality of reporting and interpretation of data.
1.Please provide a CONSORT checklist, listing the items and the page/section of the manuscript they appear.

2.State objectives at the end of the background section instead of at the end of several section.

3.How and by who was randomization done and was the allocation concealed (how)?

4.Please clearly state the exclusion criteria?

5.I recommend to specify the title to ‘a double-blinded randomized pilot trial’ as it speaks for the high quality of the study design.

6.As you have many variables of interest, please specify primary and secondary outcomes.

7.Please provide the a priori sample size calculation. If none was done, please state clearly in limitations with a statement regarding potential lack of statistical power. In this case it may be of advantage to look at effect sizes and interpret them with caution.

8. Mixed-methods – please state more clearly the way of analyzing the qualitative data

Validity of the findings

Was cognition measured and if yes how? Was there a difference in usage depending on cognitive performance? If cognition was not measured, please state this in the limitation section as the feasibility of independent use of such technology may be restricted to people with sufficient levels of cognition.

Groups differed at baseline in physical function. This could have affected the results and should be regarded in the analyses (e.g. baseline level as covariate) and discussion.

Additional comments

The revised manuscript ‘Effects of online group exercises for older adults on physical, psychological and social wellbeing: a pilot trial’ is a major improvement on the previous version and reports well and in depth about important aspects of the study. The changes improved the understanding of the intervention a lot.

Reviewer 2 ·

Basic reporting

This paper describes randomized trial, in which all participants joined an online exercise program with proven efficacy for falls prevention in older people for 8 weeks. Half of the participants joined the individual format, and half joined the group format. The aim was to investigate disparate effects of these different formats on physical, psychological and social well-being outcomes.The paper is very long for quite a simple message. The level of detail (while seemingly correct) reflects more a thesis format than a paper format. The paper would need to be shortened substantially to make it more digestible.

Experimental design

While I am not familiar with the presented app or other studies/paper by the authors, I am somewhat concerned that the authors have divided their trial outcome measures across multiple papers. If this is the case, these other papers need to be stated more clearly and a reason for the multiple papers needs to be clarified.

Also, in order to find a meaningful effect, the Otago program should be run over a minimum period of 4 to 6 months, not 2 months as in the current study. This is a limitation of the study design.

Validity of the findings

After reading the paper, it seems to me that the results are very similar across both groups and therefore both formats show identical outcomes. The study sample is small for a 2-group intervention trial. Certain statements, such as “while in the Control group fitter individuals tended to adhere more to the training, this was not the case for the Social group, where the initial level had no effect on adherence” should be rephrased and put into context of the small sample.

The app seems to have a substantial overlap with another app that I am personally more familiar with (Silveira). The authors have referenced the paper by Silveira, but a clearer description on how your app is different is warranted. For example, it seems that the current app uses an identical reward system for program using a ‘growing garden metaphor’.

Additional comments

In summary, the sample is small, the duration of the study is short and the research question is somewhat unclear.

I would like to congratulate the authors on the development of what looks a very nice app. Despite some experimental limitations as state above, the study is conducted well. However, the results need to be described in a more modest form and the paper needs to be shortened.

---

## Round 0.2 · accepted · Accept

The authors have addressed the reviewer comments and the manuscript meets the criteria for publication in PeerJ.

Reviewer 2 ·

Basic reporting

see below

Experimental design

see below

Validity of the findings

see below

Additional comments

The paper is still quite long and while my concern for fragmentation of results remains, the authors have addressed my concerns.
As a general advice for the authors for future studies: splitting up results of a trial across multiple papers is not only bad practice, it also reduces the impact of your paper.